# The F-Actin-Binding MPRIP Forms Phase-Separated Condensates and Associates with PI(4,5)P2 and Active RNA Polymerase II in the Cell Nucleus

**DOI:** 10.3390/cells10040848

**Published:** 2021-04-08

**Authors:** Can Balaban, Martin Sztacho, Michaela Blažíková, Pavel Hozák

**Affiliations:** 1Department of Biology of the Cell Nucleus, Institute of Molecular Genetics of the Czech Academy of Sciences, v.v.i., 142 20 Prague, Czech Republic; can.balaban@img.cas.cz (C.B.); martin.sztacho@img.cas.cz (M.S.); 2Light Microscopy Core Facility, Institute of Molecular Genetics of the Czech Academy of Sciences, v.v.i., 142 20 Prague, Czech Republic; michaela.blazikova@img.cas.cz

**Keywords:** MPRIP, phase separation, PIP2, actin, nucleus

## Abstract

Here, we provide evidence for the presence of Myosin phosphatase rho-interacting protein (MPRIP), an F-actin-binding protein, in the cell nucleus. The MPRIP protein binds to Phosphatidylinositol 4,5-bisphosphate (PIP2) and localizes to the nuclear speckles and nuclear lipid islets which are known to be involved in transcription. We identified MPRIP as a component of RNA Polymerase II/Nuclear Myosin 1 complex and showed that MPRIP forms phase-separated condensates which are able to bind nuclear F-actin fibers. Notably, the fibrous MPRIP preserves its liquid-like properties and reforms the spherical shaped condensates when F-actin is disassembled. Moreover, we show that the phase separation of MPRIP is driven by its long intrinsically disordered region at the C-terminus. We propose that the PIP2/MPRIP association might contribute to the regulation of RNAPII transcription via phase separation and nuclear actin polymerization.

## 1. Introduction

Myosin phosphatase rho-interacting protein (MPRIP) was first described as a cytoskeletal protein involved in the regulation of stress fibers [1,2,3]. Recently, we identified MPRIP by label-free quantitative mass spectrometry as a nuclear Phosphatidylinositol 4,5-bisphosphate (PIP2) interactor [4]. Its depletion was reported to stabilize and thus increase the number of actin stress fibers in smooth muscle cells, whereas its overexpression leads to the disassembly of stress fibers in neuronal cells [5,6]. Its binding to F-actin stress fibers is mediated by the N-terminal region which also contains two Pleckstrin Homology (PH) domains [7,8]. These PH domains possess a lipid binding site where the positively charged residues (i.e., Lysine and Arginine) enable the binding of the negatively charged inositol head group of the phosphatidylinositol phosphates (PIPs) [9].

It was shown that the PH-like domains are responsible for the localization of proteins such as Nuclear Myosin I (NM1) to nuclear PIP2 [10,11,12]. Recently, we proposed a model where PIP2 localizes into discrete nuclear areas where it regulates processes such as transcription and splicing [12]. Moreover, we described the importance of nanoscale, nucleoplasmic PIP2-rich structures—nuclear lipid islets (NLIs) in RNA polymerase II (RNAPII) mediated transcription and showed that their surface is in the proximity of chromatin, RNA and NM1 [12,13,14]. However, the precise mechanism of interplay among PIP2, NM1 and RNAPII in regulation of transcription remains elusive. Recently, it was shown that the activity of RNAPII is regulated by its propensity to phase separate [15,16]. The phase separation of RNAPII is regulated through the phosphorylation of its intrinsically disordered C-terminal domain (CTD) [17]. In the last decade, it became evident that the intrinsically disordered regions (IDRs) are responsible for the regulation of diverse nuclear processes through functional compartmentalization into membraneless nuclear bodies [18,19,20,21]. The intermolecular multivalent interactions of IDRs lead to the formation of condensates with liquid-like properties [22,23,24]. The liquid–liquid phase separation (LLPS) is an entropy-driven mechanism which provides the basis for the biophysical explanation of the compartmentalization of biological processes in the dense nuclear environment [25,26].

In this study, we show that MPRIP localizes to PIP2-containing nuclear structures, forms complex with RNAPII and MYO1C-an isoform of NM1-and undergoes LLPS (liquid–liquid phase-separation) in cell nucleus while preserving its F-actin binding capacity [27,28]. Furthermore, we describe the liquid-like properties of MPRIP protein in the cell nucleus. Therefore, we hypothesized that MPRIP might represent the functional hinge in the transcriptional regulation of RNAPII compartmentalization through actin, PIP2 and NM1 interaction.

## 2. Material and Methods

### 2.1. Cell Cultures and Transfections

Human cervical carcinoma (HeLa, ATCC no. CCL2) cells and human osteosarcoma (U2OS, ATCC no. HTB96) cells were grown in Dulbecco’s modified Eagle’s medium (DMEM, Sigma D6429) with 10% FBS at 37 °C in a humidified 5% CO_2_ atmosphere. HeLa cells in suspension were kept in minimum essential medium Eagle with Spinner modification (S-MEM, Sigma M8167) supplemented with 5% FBS at 37 °C in a humidified 5% CO_2_ atmosphere. Transfections were carried out using Lipofectamine 3000 (Invitrogen, Thermo Fisher Scientific, Waltham, MA, USA) according to the manufacturer’s protocol. Stable cell lines were established by sorting the transiently transfected cells and keeping them under selective media (G418, Sigma G8168). The cells were then evaluated by Western blot and fluorescence light microscopy (Appendix A).

### 2.2. Constructs and Antibodies

pDEST53 GFP-MPRIP (Human Isoform 3 of MPRIP) plasmid was used for the overexpression of MPRIP in U2OS cells (Appendix A). The N-terminal (1–450th amino acid) and C-terminal (450–1000th amino acid) regions of MPRIP were amplified by PCR using pDest-53-GFP-MPRIP as a template. The N-terminal region was inserted in pEGFP-N1, while the C-terminal region was inserted in pEGFP-C1. The constructs were transiently overexpressed in U2OS cells to examine their specific localization and features. For the detailed structure of the fragment, please see Appendix A:Anti-MPRIP antibody-HPA022901 (SigmaAldrich, St. Louis, MO, USA)Anti-PI (4,5) P2 antibody: Z-A045, clone 2C11 (Echelon, San Jose, CA, USA)Anti-RNAPII CTD Phospho S5 antibody-ab5131 (Abcam, Cambridge, UK)Anti-Lamin B1 antibody—ab16048 (Abcam, Cambridge, UK)Anti-GAPDH antibody [6C5]-ab8245 (Abcam, Cambridge, UK)Rabbit Anti-Mouse Immunoglobulin G H&L—ab46540 (Abcam, Cambridge, UK)Anti-MYO1C antibody was supplied by Peter G. Gillespie, Oregon Hearing Research Center and Vollum Institute [29].

### 2.3. Bioinformatics Analyses

Nuclear Localization Signal (NLS) prediction was done by using the online tool cNLS Mapper (Appendix A) [30]. The Clustal Omega test, analyzing the degree of the conserved domains, was performed by the tool provided by European Bioinformatics Institute (Appendix A) [31]. The IDRs of the MPRIP were predicted by the D2P2 tool; Database of Disordered Protein Predictions (Appendix A) [32].

### 2.4. Immunofluorescence Labelling

U2OS cells grown on high-performance cover glasses of 12 mm in diameter with restricted thickness-related tolerance (depth = 0.17 mm ± 0.005 mm) and the refractive index = 1.5255 ± 0.0015 (Marienfeld 0107222). The cells were fixed with 4% formaldehyde for 20 min and permeabilized with 0.1% Triton X-100 for 5 min. Non-specific binding was blocked by 5% Bovine Serum Albumin (BSA). All solutions were diluted in Phosphate-Buffered Saline (PBS). Then, the cells were incubated with anti-PtdIns(4,5)P2 and anti-MPRIP antibodies, and mounted in 90% glycerol with 4% n-Propyl gallate (NPG). For the super-resolution microscopy, five 5 min washes in PBS with 0.1%Tween were made between each step.

### 2.5. Stimulated Emission Depletion (STED) Microscopy

The images were acquired using a Leica TCS SP8 STED 3× microscope equipped with a Leica DFC365 FX digital camera with a STED white CS 100 × 1.40 NA oil objective for the optimized overlay of excitation and an STED beam (Leica Mikrosysteme Vertrieb GmbH, Wetzlar, Germany). Image capturing was performed using the Leica LAS × 64-bit software package. The acquired images were deconvolved using Huygens Professional software version 19.04 (Scientific Volume Imaging, Hilversum, Netherlands), using the Classic Maximum Likelihood Estimation (CMLE) algorithm, with SNR:07 and 20 iterations. The image analysis was carried out using the Coloc 2 plugin of Fiji software. The significance of each statistical analysis was determined by a Student’s t test. The randomized images for statistical tests were obtained by rotating one of the two channels 90 degrees [33]. The resolution format was taken at 1024 × 1024 and the corresponding pixel sizes were both 28 nm in x and y. A minimum of five cells were analyzed per each condition.

### 2.6. Nuclear Extraction and Pull-Down Assay

Nuclear lysate fraction was prepared from the suspended HeLa cells as described by Trinkle-Mulcahy [34]. Cytoplasmic fraction was obtained by Dounce homogenization (Kontes 885300-0015) on ice (30 strokes), following three cold PBS washes. The homogenate was incubated 20 min on ice in buffer 150 mM NaCl, 1% NP-40, 50 mM Tris pH 7.4, 1 mM DTT supplemented with protease inhibitors from F. Hoffmann (La Roche Ltd., Basel, Switzerland, 05056489001) followed by centrifugation at 4000× *g* at 4 °C. The nuclear pellet was sonicated at Soniprep 150 (MSE) bench top sonicator (1 sec on, 1 s off for 30 cycles at power 10 amplitude microns). Sonicated lysate was spun down at 13,000× *g* for 15 min at 4 °C. Supernatant was collected as nuclear lysate. Protein concentration was determined by Pierce™ BCA Protein Assay (Thermo Scientific, 23227) according to the manufacturer’s protocol.

In pull-down experiments, 3 mg of HeLa total nuclear lysate was used for each condition. Fifty µL of PIP-coated beads slurry were added to each condition and incubated overnight in the nuclear lysate. The beads were washed five times with 1 mL of ice-cold nuclear extract buffer (150 mM NaCl, 1% NP 40, 50 mM Tris pH 7.4, 1 mM DTT, protease inhibitors from F. Hoffmann (La Roche Ltd., Basel, Switzerland, 05056489001). Two times Laemmli buffer was added to the beads, boiled for 5 min and loaded to the SDS gel followed by Western blot analysis. The PIP-coated beads were obtained from Echelon Biosciences Inc., Salt Lake City, UT, USA (Control Beads, P-B000; PI(3)P beads, P-B003A; PI(4)P beads, P-B004A; PI(5)P beads, P-B005A; PI(3,4)P2 beads, P-B034A; PI(3,5)P2 beads, P-B035A; PI(4,5)P2 beads, P-B045A; PI(3,4,5)P3 beads, P-B345A).

### 2.7. Co-Immunoprecipitation Assay

Nuclei from suspended HeLa cells were prepared as described previously [34]. One milligram (1 mg) of nuclear lysate was incubated with 2 µg MPRIP antibody at 4 °C overnight. The lysates were incubated with G-protein magnetic beads (Pierce™, Thermo Fischer 88848) at 4 °C for 1 h. Beads were washed 5× times by buffer (150 mM NaCl, 1.0% NP-40, 50 mM Tris pH 7.4, 1 mM DTT) and subjected to Western blot analysis.

### 2.8. Live-Cell Imaging Microscopy

U2OS cells expressing pDEST53-GFP-MPRIP were grown on 35 mm glass bottom dishes (# P35G-1.5-14-C, MatTek Corporation, Ashland, OR, USA). Live-cell imaging was performed using a Leica TCS SP5 Confocal microscope equipped by an environmental chamber with CO_2_ and temperature control. Image acquisition started 24 h post transfection. For the video (Video S1), the cells were recorded every 5 min for 8 h and 10 min, starting from the 12th hour after transfection. The video has a pixel size of 197 nm × 197 nm.

### 2.9. Hexanediol Treatment

Twenty-four hours after transfection, the cells were exposed to 25% aliphatic alcohols dissolved in PBS (1,6-Hexanediol and 2,5-Hexanediol, SigmaAldrich, St. Louis, MO, USA, Cat. No. 240117, H11904) for two minutes at room temperature. GFP signals of the nuclear droplets were recorded by Leica TCS SP8 confocal microscope for 2 min.

### 2.10. FRAP

Cells were transfected a day before the experiment on 35 mm coverslip bottom dishes (catalogue # P35G-1.5-14-C, MatTek Corporation, Ashland, MA, USA). Fluorescence recovery after photobleaching (FRAP) experiments were performed on Leica TCS SP8 confocal microscope with Leica HC PL APO 63×/1.40 oil CS2 objective. Scanning speed was set to 1800 Hz with bidirectional X and an imaging acquisition rate was 0.0232 s Ten pre-bleach images were captured and the photo bleach was performed for another 10 frames and finally, one thousand post-bleach images were acquired. Full-and inner-FRAPs were performed more than one hundred times. The fiber-FRAP was performed at least 50 times and ten bleaches were done for nucleoplasmic-FRAP. Solidified structures were bleached as full-FRAP experiments (*n* = 20).

Protein mobility within the condensate was measured by quantifying the recovery of the bleached area at the coast of the unbleached region by a custom written MATLAB script (ver. R2019b, The MathWorks, Inc., Natick, MA, USA). The recovery curves were fitted with a single and/or bi-exponential curve, using the non-linear least squares (lsqcurvefit) function. The bleached region was corrected for general bleaching during image acquisition.

We employed FRAP on the circular regions of four different sizes. Full-FRAP experiments covered the area of the whole condensate which was around 3 µm in diameter. In inner-FRAP experiments, the bleached area was a circle of 1 µm in diameter within the 3 µm condensates. Nucleoplasmic-FRAP was performed on a circular area of 6 µm in diameter. For the fiber-FRAP, the bleached are was taken as 0.5 µm in diameter. All FRAP experiments except fiber-FRAP were fitted to a single exponential curve; the fiber-FRAP was fitted to the bi-exponential curve.

### 2.11. Phalloidin Staining of GFP-MPRIP Expressing U2OS Cells

Cells were transfected with pDEST53-GFP-MPRIP on coverslips and fixed by 4% formaldehyde. F-actin was visualized by 1.5 µg/mL Alexa Fluor™ 568 Phalloidin (Invitrogen, Carlsbad, CA, USA, catalogue # A12380) staining following a standard indirect immunofluorescence labelling protocol. Images were acquired by the STED microscopy as mentioned in Section 2.5. The resolution format was taken as 256 × 256 and the corresponding pixel sizes were both 13 nm in x and y.

## 3. Results

### 3.1. MPRIP Protein Is Present in the Cell Nucleus

The MPRIP protein was previously described as a cytoplasmic protein localizing to F-actin stress fibers [1,2,5]. It was also identified as MYO1C interactor by quantitative mass spectrometry (qMS) [35]. Therefore, this protein might represent a link between nuclear actin and the transcriptional factor NM1, suggesting a role in the RNAPII transcription process. To evaluate if MPRIP has a nuclear function, we first investigated its nuclear localization. We performed an indirect immunofluorescence (IF) labelling on U2OS cells using MPRIP-specific antibody.

The MPRIP IF labelling confirmed the previously described cytoplasmic stress fiber localization (Figure 1A). Moreover, we observed that MPRIP localizes inside the nucleus, displaying a granular pattern that was dispersed in the nucleoplasm (Figure 1A). In order to confirm its nuclear presence, we analyzed the fractionated lysates of HeLa cells. The Western blot (WB) analysis of nuclear and cytoplasmic fractions confirmed that MPRIP occurs in both environments (Figure 1B). Lamin B and GAPDH were used as the purity controls of fractions. The nuclear fraction was devoid of GAPDH, whereas Lamin B was detected exclusively in nuclear fraction (Figure 1B).

This protein of ~120 kDa would require an NLS region to be targeted to the nucleus. Our bioinformatics analysis predicts a NLS region at residues 155 to 164 (Appendix A). Clustal Omega test run on eight different MPRIP sequences (three isoforms in Homo sapiens and five orthologues from other mammals) delineated that the predicted NLS region and the PH domains were highly conserved (Appendix A).

### 3.2. Nuclear MPRIP Interacts with PIP2 and Forms a Complex with RNAPII and MYO1C

We recently identified PIP2 as an important player in the regulation of RNAPII transcription, presumably through NM1 interaction [12]. Therefore, we next sought to determine whether PH domain-containing MPRIP has the capacity to bind PIPs [36]. We used PIP-covered beads in pull-down experiments in nuclear lysates and detected the bound MPRIP protein by WB (Figure 2). These experiments showed that MPRIP interacts specifically with PI4,5P2. Thus, we asked whether MPRIP localizes to the proximity of PIP2 in the cell nucleus.

We stained the U2OS cells with MPRIP-and PIP2-specific antibodies and revealed their sub-nuclear localization by STED microscopy. The nuclear PIP2 localizes to three different compartments: nuclear speckles, NLIs and nucleoli [12,37]. The statistical analysis revealed that MPRIP localizes to the proximity of the PIP2 signal in the nuclear speckles and NLIs (Figure 3). Two statistical coefficients, Manders and Spearman, were determined for two different PIP2-rich compartments, where all tests scored significant differences compared to the randomized images (Figure 3D,G).

Our data presented here indicate that MPRIP colocalizes with nucleoplasmic PIP2-rich NLI structures which were shown to be sites of RNAPII active transcription [12]. Therefore, we suggest that MPRIP might represent the possible link between RNAPII and transcription factor NM1, a MYO1C isoform. Therefore, we tested whether MPRIP associates with RNAPII and MYO1C by MPRIP immunoprecipitation (IP) followed by WB (Figure 4). Our data show that MPRIP is in the same complex with MYO1C and the active form of RNAPII (phosphor Ser5) which was previously shown to associate with PIP2-rich NLIs [12]. Thus, it is possible that MPRIP is involved in an ongoing transcription process that occurs at the surface of NLIs.

### 3.3. Overexpression of MPRIP Leads to Formation of Condensates and Fibrous Structures in the Cell Nucleus

To further study the nuclear localizations of MPRIP, we overexpressed GFP-tagged MPRIP in human U2OS cells. Interestingly, the transfected cells showed three different phenotypes (Figure 5A–C). The first and most common (~70%) phenotype recapitulates the pattern of IF labelling of endogenous MPRIP which is fine granular foci that is diffused throughout the nucleoplasm (Figure 5A). The second phenotype resembles globular condensed structures in the nucleoplasm, which had 1 to 5 µm in diameter (Figure 5B). The third phenotype of the MPRIP-expressing cells shows nucleoplasmic, fibrous structures that are intertwined (Figure 5C).

The large globular condensates started to appear 8 h post-transfection and persisted throughout 72 h after transfection. The fibrous structures started to appear mostly at 20th hour of post transfection. Interestingly, the population of cells forming large condensates and fibrous structures increased over time after transfection, whereas the population of cells with a diffused pattern decreased (Figure 5D).

### 3.4. Nuclear MPRIP Condensates Are Formed by Phase Separation

Our data show that the overexpression of MPRIP leads to the formation of spherical condensates. We speculated that the formation of these roundish structures might be driven by LLPS. Therefore, we utilized the aliphatic alcohol 1,6-hexanediol, which is widely used for the disruption of weak hydrophobic interactions driving phase separation [38,39]. In contrast, the derivative 2,5-Hexanediol has minimal impact on the phase separated condensates and thus serves as a negative control [40,41,42]. To demonstrate the LLPS ability of condensed GFP-MPRIP structures, we treated the cells with 25% of 1,6- and 2,5-Hexanediol/PBS solution for 2 min (Figure 6A). The MPRIP condensates started to dissolve 10 s after the addition of 1,6-Hexanediol and disappeared completely after 2 min of incubation. On the contrary, the treatment of 2,5-Hexanediol did not lead to the complete dissolution of the condensates, even after 2 min of incubation (Figure 6B).

### 3.5. Dynamic Liquid-Like MPRIP Condensates Are Able to Fuse, Segregate and Form Fibers

The phase-separated structures are usually spherical and they show dynamic behavior with a capacity to fuse and split [26,43]. In our live-cell imaging experiments, we observed that GFP-MPRIP condensates show dynamic motility; they fuse and form larger condensates, preserving their spherical shape (Figure 7A) and finally they divide (Video S1, 0:55 to 1 h). Figure 7B represents images from the video S1 that shows the ability of GFP-MPRIP condensates to transform into the fibers and then back into condensates.

### 3.6. MPRIP Condensates Show High Internal Dynamics and Rapid Molecular Interchange between Condensates and Neighboring Nucleoplasm

To further study the liquid-like properties of MPRIP condensates, we measured their molecular dynamics by FRAP experiments. For this purpose, we performed a total of five FRAP experiments corresponding to each phase and structure. We started by full-FRAP experiment in order to assess the rate of molecular exchange between nucleoplasm and MPRIP condensed structures (Figure 8A). The half-time was recorded as 3.6 (±1.2) s with a diffusion coefficient D of 0.147 µ^2^/s. As described by Patel et al. in 2015, we also observed a solidified, gel-like phase of the condensates [44]. This phase occurs more frequently with increasing time after transfection. In the first 24 h post transfection, the mobile fraction of the full-FRAP was ~60%, which dropped down to ~10% in 48 h post transfection (Figure 8B, solidified structure).

In order to confirm the liquid-like properties, it was necessary to measure the inner dynamics of MRPIP condensates (Figure 8C, inner-FRAP). The measured half-time of the recovery was 1.6 (±0.6) seconds with 85% mobile fraction (Figure 8D). To determine the mobile fraction of the fibers, the nucleoplasmic mobile fraction needed to be measured first. This was due to the small diameter of the fiber that causes the nucleoplasm to interfere with the calculations of the fiber-FRAP. The nucleoplasm was bleached as a circle with 3 µm radius and the calculated half-time of the recovery was around 0.5 (±0.15) seconds (Figure 8C). Our data indicate that the nucleoplasmic GFP-MPRIP moves by free diffusion, which is determined by the mobile fraction ~96% (Figure 8D). The fiber-FRAP was performed and the half-times were determined as follows: fast fraction = 0.2 (±0.15) seconds and slow fraction = 1.6 s. The data show two fractions. One corresponds to the free diffusion (nucleoplasmic-FRAP) and the second corresponds to the fiber itself. The mobile fraction of the fiber-FRAP was 45% (Figure 8D).

Interestingly, the inner fraction of a fiber shows the same dynamics as the inner fraction of a condensate. However, the difference in the mobile fractions (inner-FRAP: 85%, fiber-FRAP: 45%) suggests that 40% of the immobile fraction is bound to an additional factor; presumably, to nuclear actin (see below) (Figure 8D).

### 3.7. MPRIP Fibers Contain Nuclear F-Actin

The results of FRAP experiments prompted us to further investigate the fibrous structures which presented an immobile fraction. Since MRPIP is an F-actin binding protein, we tested whether F-actin might be responsible for the immobile fraction of the fibrous MPRIP structures [1]. To examine this process, GFP-MPRIP was overexpressed in U2OS cells for 24 h and were subsequently stained with phalloidin. Indeed, we observed that the actin fibers labelled by phalloidin were decorated by GFP-MRPIP (Figure 9A,B). This suggests that condensed MPRIP protein retains the F-binding capacity in nuclear environment. In Figure 9A, the GFP-MPRIP condensates are aligned on fibers adopting an elongated shape that extrude thin, connecting lines localizing to the phalloidin labelled fibers. Furthermore, STED microscopy was employed on the cells presenting only GFP-MPRIP fibers (without the globular condensates) to visualize the structure in more detail that determined its thickness ranging from 150 to 250 nm (Figure 9B).

### 3.8. The C-Terminal IDR Domain Is Responsible for Phase Separation of MPRIP

To determine the regions that are responsible for MPRIP propensity for phase separation and fiber formation, we split MPRIP into two fragments. We prepared the GFP tagged constructs which comprise different domains (see Materials and Methods 2.2). The first fragment F1 consists of the N-terminal domain of MPRIP; carrying F-actin binding and the NLS region. The fragment F2 consists of the C-terminal domain, comprising a long IDR with regions that are known to interact with F-actin, regulating proteins such as PP1, MYPT1 and RhoA [8]. The fragments overexpressed in U2OS cells show a different cellular localization (Figure 10). The F1 fragment contains NLS, thus mainly localizing to the nucleus. However, it also binds to the F-actin stress fibers (Figure 10A). Moreover, the F1 fragment shows no sign of phase separation. The fragment F2 localizes only to the cytoplasm where it forms fibrous and globular structures reminiscent of those formed by phase separation (Figure 10C). Interestingly, we also noticed that the F2 fragment was also able to bind to the F-actin cytoskeleton (Appendix A).

## 4. Discussion

PIP2, as the integral component of the cytoplasmic inner membrane leaf, is a crucial regulator of F-actin polymerization [44,45,46]. PIP2 regulates actin cortex, membrane tension, raft formation, endocytosis, motility and forms signaling hubs [47,48,49,50]. In the nucleus, PIP2 is an integral element of nuclear compartments which are involved in transcription, epigenetic regulation, the enzymatic activity of fibrillarin and presumably, mRNA splicing [12,13,51,52,53,54]. Recently, the nuclear PIP2 interactors were identified by an advanced quantitative mass spectrometry approach and showed that the polybasic K/R regions which are responsible for PIP2 interaction are a common feature of PIP2-effectors [4,55].

Initially, we showed that MPRIP is a nuclear protein that exists in two forms (Figure 1B) which presumably represent different phosphorylation states due to a number of predicted phosphorylation sites (Appendix A). Our super-resolution microscopy followed by the statistical analysis of MPRIP spatial distribution showed that MPRIP localizes predominantly to PIP2-rich nuclear speckles and NLIs (Figure 3D,G). NLIs were previously shown to be important for an active RNAPII transcription [12]. The PIP2 nuclear localization was visualized in unprecedented resolution using advanced super-resolution microscopy—direct stochastic optical reconstruction microscopy (dSTORM)— at nuclear speckles and NLIs where PIP2 was found in close proximity with the subset of RNAPII [56]. Furthermore, we showed that the PH domain-containing MPRIP protein binds PIP2 in vitro and thus suggest the importance of PIP2 for its activity. These data suggest that MPRIP is a nuclear protein whose nuclear localization presumably depends on the PIP2.

Our immunoprecipitation experiments determined the unconventional myosin- MYO1C and the active RNAPII as the nuclear interactors of MPRIP (Figure 4). MYO1C possess a C-terminal, PH-like domain which enables its binding to PIP2 [11,57,58]. An isoform of MYO1C termed NM1 was previously shown to be important for RNAPI and RNAPII mediated transcription [12,37,59]. Moreover, NM1 was defined as a transcription factor with the capacity to bind nuclear actin; a component of RNAPII pre-initiation complex and an important RNAPI regulator [59,60]. These data indicate a potential function for MPRIP in transcription regulation.

The overexpression of MPRIP induces the formation of granular nucleoplasmic foci and the spherical condensates depending on the time after transfection (Figure 5A,B,D). The 1,6-hexandiol treatment showed that the condensates are formed by the phase separation (Figure 6) and that these condensates display liquid-like properties (Figure 7A and video S1) [24,43]. Further, we observed the formation of MPRIP fibers in the cell nucleus (Figure 5C). We monitored these fibers for several hours and observed reformation of spherical condensates when fibers disassembled (Figure 7B, Video S1). To the best of our knowledge, this is a novel behavior for phase-separated proteins that shows liquid-like properties even when bound to a fibrous structures (Figure 9). Our FRAP data determined high mobility for MPRIP molecules within both structures—condensates and fibers (Figure 8). MPRIP condensates showed time-dependent solidification, as another intrinsic property of phase-separating proteins [24,61]. Moreover, the FRAP experiments showed that fibers contain a larger immobile fraction compare to spherical condensates. We assume that this immobile fraction indeed represents the MPRIP molecules bound to nuclear F-actin (Figure 9).

The N-terminus of MPRIP contains the F-acting binding domain that covers one of the PH domains, the NLS region and the short IDR region interspaced between both PH domains [8]. In Figure 10, we showed that the N-terminus region is responsible for MPRIP protein localization to the nucleus but not for its capacity to phase separate (Appendix A). Due to the overlap of PH and acting binding domains, we hypothesized that the binding of actin to MPRIP regulate PIP2-binding capacity and thus affects its nuclear localization. The impact of cytosolic/nuclear actin concentration equilibrium onto the MPRIP function remains elusive.

We showed that the C-terminal part of MPRIP is responsible for its phase separation. Interestingly, this fragment also showed to localize to cytosolic F-actin (Appendix A). This is contrary to in vitro observations of other authors showing that the C-terminal region is not responsible for a direct F-actin binding. However, it was shown to interact with actin regulators such as RhoA, MYPT1 and NUAK2 [2,3,6]. Therefore, we speculate that the C-terminal F-actin binding (Appendix A) and fiber formation (Figure 10) is mediated indirectly through these factors.

Nuclear actin is an enigmatic factor which was identified as an interactor of a plethora of nuclear proteins such as transcriptional factors [11,62,63,64,65]. However, the nuclear F-actin formation is shown to be crucial for the relocalization of heterochromatin breaks and were associated with chromatin-modifying enzymes such as deacetylases, acetyltransferases and DNA repair factors [66,67]. Moreover, the formation of nuclear actin fibers is associated with defects in nuclear topology and RNAPII relocalization [68]. We observed the production of the nuclear MPRIP fibers when the formation of nuclear F-actin was stimulated by heat shock (Appendix A) [1]. Therefore, the formed nuclear F-actin might attract MPRIP and thus alter the RNAPII transcription.

In conclusion, we provided evidence for the presence of MPRIP, an actin regulatory protein, in cell nucleus. The MPRIP protein localizes to nuclear PIP2-rich sub-compartments and it was identified as a component of the RNAPII/NM1 complex. The expression of this protein leads to the formation of phase-separated condensates that are able to bind nuclear F-actin fibers. We showed that this binding is reversible and the spherical condensates can reform when fibers disappear (summary in Figure 11). These data suggest that the PIP2-associated RNAPII transcription might be regulated by phase separation that is dependent on nuclear actin polymerization.

## Figures and Tables

**Figure 1 cells-10-00848-f001:**
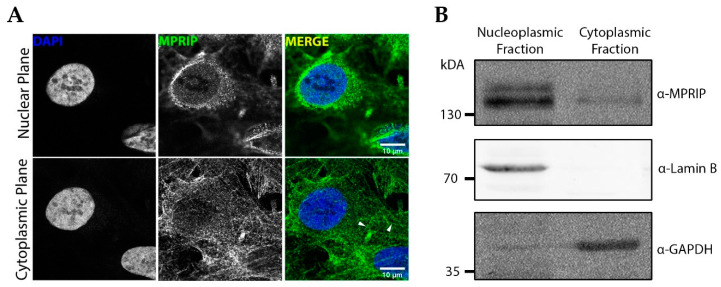
Myosin phosphatase rho-interacting protein (MPRIP) localizes to the cytoplasm and cell nucleus: (**A**) confocal microscopy of an immunofluorescence (IF) experiment shows the localization of endogenous MPRIP in U2OS nucleus and cytoplasmic stress fibers. Upper and lower images represent different focal planes: nuclear and cytoplasmic, respectively. Arrowheads point to the MPRIP staining on stress fibers. Scale bars represent 10 µm; (**B**) Western blot (WB) of HeLa cells fractionated to nuclear and cytoplasmic fractions with detected MPRIP, Lamin B and GAPDH by a specific antibody (*n* = 3).

**Figure 2 cells-10-00848-f002:**
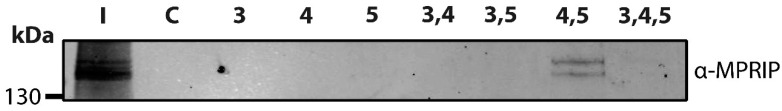
MPRIP specifically interacts with Phosphatidylinositol 4,5-bisphosphate (PIP2). Various PIP-coated agarose beads were used as a bait to pull-down interacting proteins from HeLa nuclear fraction. The membrane was WB analyzed with anti-MPRIP antibody. I—input 2%; C—control beads; 3—PI(3)P, 4—PI(4)P; 5—PI(5)P; 3,4—PI(3,4)P2; 3,5—PI(3,5)P2; 4,5—PI(4,5)P2; 3,4,5—PI(3,4,5)P3. *n* = 3.

**Figure 3 cells-10-00848-f003:**
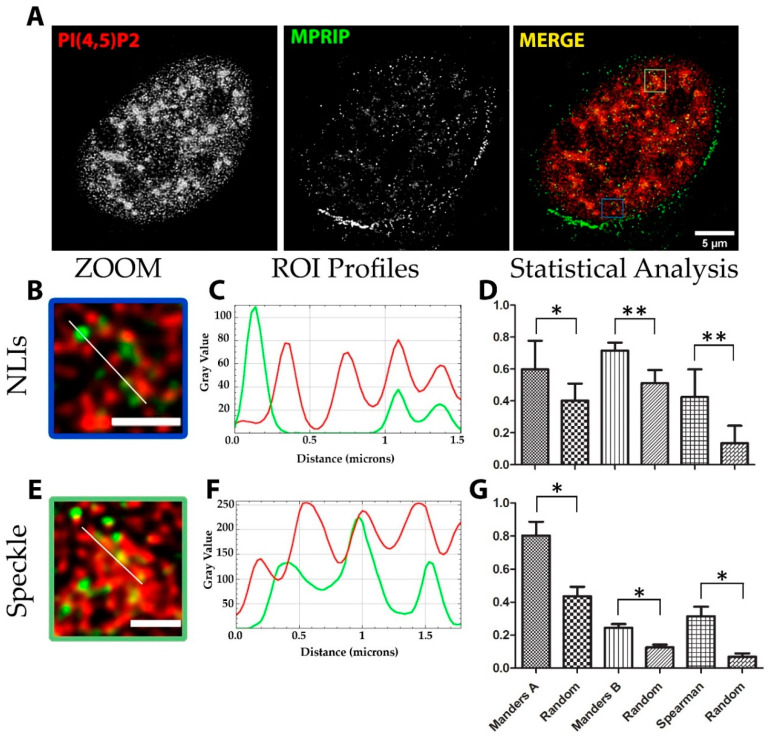
Super-resolution microscopy revealed that Myosin phosphatase rho-interacting protein (MPRIP) localizes in close proximity to Phosphatidylinositol 4,5-bisphosphate (PIP2) at nuclear speckles and nuclear lipid islets (NLIs): (**A**) immunofluorescence labelling experiment showing the presence of endogenous MPRIP and PIP2 in U2OS nucleus. Scale bar represents 5 µm; (**B**–**D**) zoomed view, intensity profiles and statistical analysis of the NLIs. Scale bar corresponds to 1 µm. (**E**–**G**) Zoomed view, intensity profiles and statistical analysis of the speckle-associated PIP2 pool. Scale bar corresponds to 1 µm. Intensity profiles plotted correspond to the lines on the zoomed view. (**D**,**G**) The bar graph shows the statistical Mander’s and Spearman coefficients of the MPRIP and PIP2 signal. Mander’s A analysis: PIP2 over MPRIP channel, and Mander’s B analysis: MPRIP over PIP2 channel. Bars that are marked as Random were obtained by analyzing randomized images (see M.M.). Single asterisk * corresponds to a significance level of *p* ≤ 0.05 and the double asterisks ** to *p* ≤ 0.007.

**Figure 4 cells-10-00848-f004:**
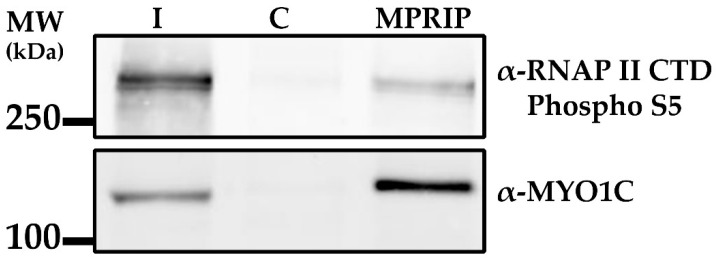
The immunoprecipitation assay confirms the existence of a MPRIP/RNAPII/MYO1C nuclear complex. The MPRIP immunoprecipitation was performed with HeLa nuclear lysates and the corresponding proteins were detected by Anti-RNAP II CTD Phospho S5 and anti-MYO1C antibodies. I—input 2%; C—control non-specific rabbit Immunoglobulin G. *n* = 3.

**Figure 5 cells-10-00848-f005:**
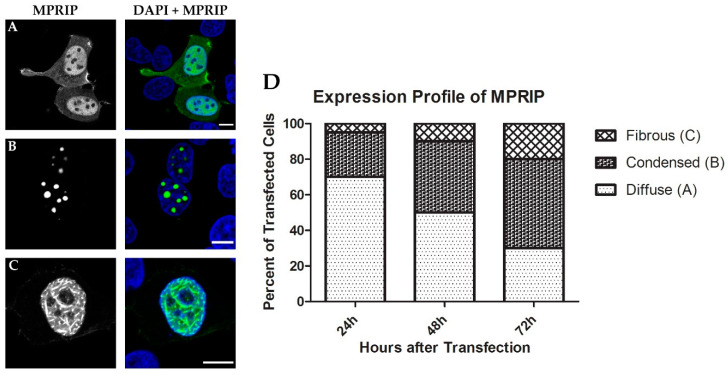
The overexpression of GFP-MPRIP results in three different phenotypes. Confocal microscopy of U2OS cells over-expressing GFP-MPRIP. (**A**) diffuse; (**B**) condensed; and (**C**) fibrous structures, which constitute the three different phenotypes. DAPI in blue and MPRIP in green. Scale bars correspond to 5 µm. (**D**) The graph reflects the changes in population of cells showing different phenotypes over the time post transfection.

**Figure 6 cells-10-00848-f006:**
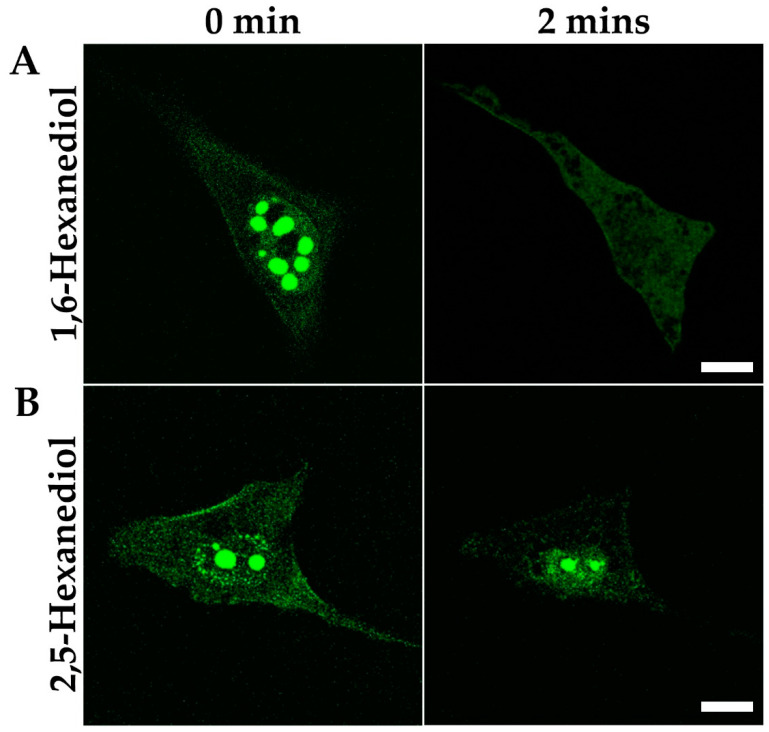
1,6-Hexanediol treatment of the cells overexpressing GFP-MPRIP dissolves the condensates. Live cell imaging showing U2OS cells over-expressing GFP-MPRIP before and after 2 min of Hexanediol treatment: (**A**) the GFP-MPRIP condensates in a cell nucleus treated with 1,6-Hexanediol; (**B**) with 2,5-Hexanediol. Green is GFP-MPRIP. Scale bars correspond to 10 µm. *n* = 2.

**Figure 7 cells-10-00848-f007:**
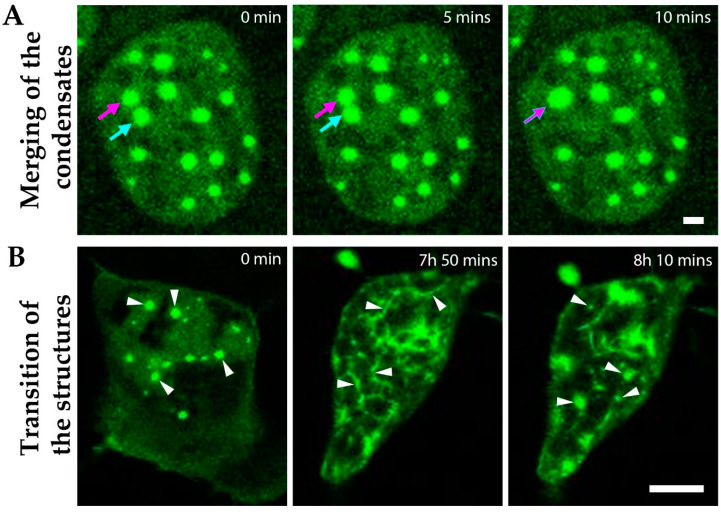
MPRIP condensates show typical features of liquid-like structures. Live cell imaging showing the nucleus of an U2OS cell over-expressing GFP-MPRIP. (**A**) The GFP-MPRIP condensates were visualized for 15 min. Arrows show the structures that merge during the imaging. Green is GFP-MPRIP. Scale bar represents 3 µm. (**B**) The GFP-MPRIP expressing cell was visualized for 8 h and 10 min. Arrowheads show the spherical condensates that disappear and form fibers, and then revert back into condensates over time. Imaging started at the 12th hour after transfection. The scale bar represents 10 µm.

**Figure 8 cells-10-00848-f008:**
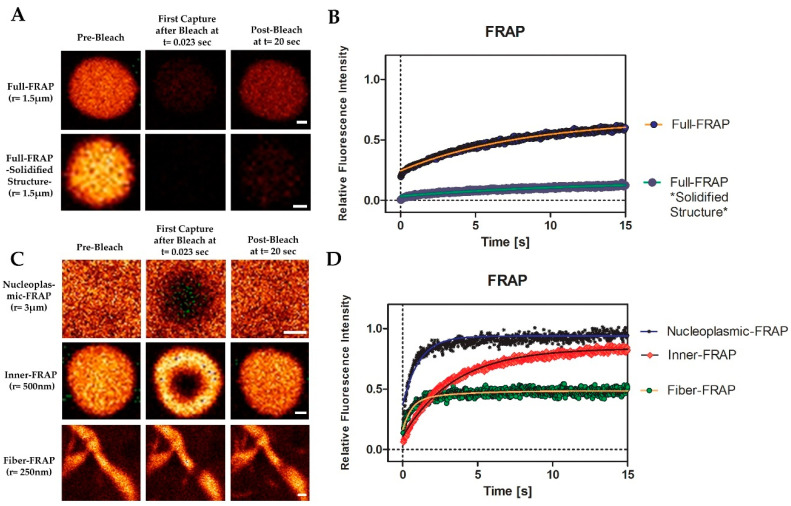
MRPIP nuclear condensates show dynamic liquid-like properties. (**A**) Representative images of Full-Fluorescence recovery after photobleaching (FRAP) experiments that were performed on GFP-MPRIP nuclear condensates. The lower row images correspond to a solidified structure. Imaging rate was 23 ms. Scale bars represent 0.5 µm. (**C**) Representative images of nucleoplasmic-, inner- and fiber-FRAP. r is the radius of the circular bleached area. Scale bars represent 3 µm, 0.5 µm and 0.25 µm from the top to the bottom, respectively. (**B**–**D**) The graphs represent the relative fluorescence values in time, starting with the first capture after bleaching. Each point corresponds to the corrected and normalized fluorescence values of the bleached areas of the FRAP experiments. The lines passing through the points are single exponential curves except for fiber-FRAP. The line for fiber-FRAP is a bi-exponential curve. The FRAP graph was created using one exemplary experiment for each FRAP experiment. “*” in subfigure B corresponds to a solidified structure.

**Figure 9 cells-10-00848-f009:**
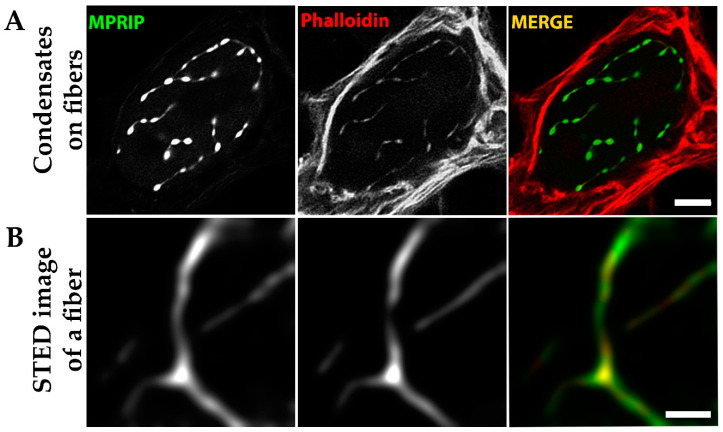
Phalloidin staining of the GFP-MPRIP overexpressing U2OS cells. Fixed GFP-MPRIP transfected cells were stained with phalloidin and investigated by confocal microscopy. (**A**) GFP-MPRIP condensates localized on phalloidin labelled fibers in cell nucleus. Scale bar represents 5 µm (**B**) GFP-MPRIP fibers visualized by STED microscopy. Blue is DAPI, green is GFP-MPRIP and red is the phalloidin staining. The scale bar represents 1 µm.

**Figure 10 cells-10-00848-f010:**
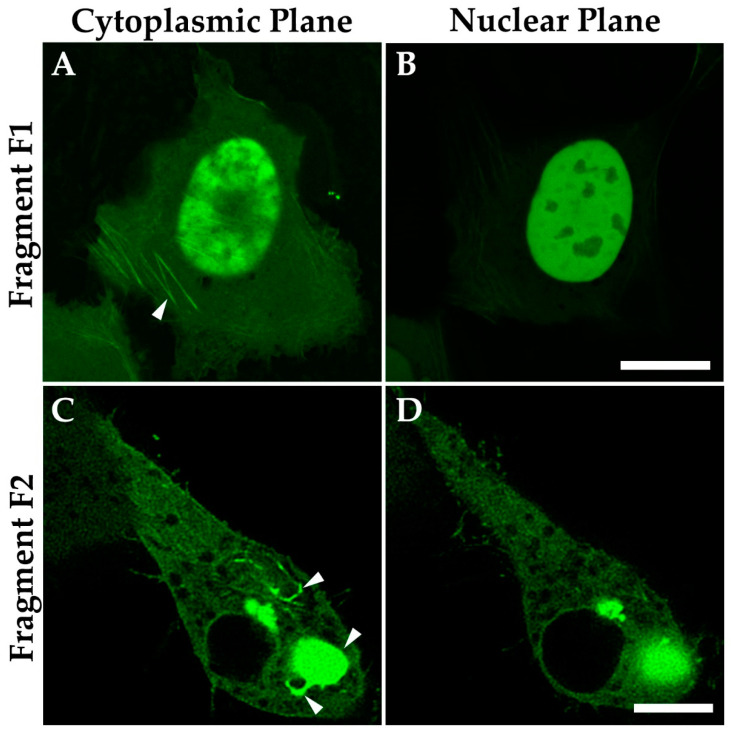
Confocal microscopy revealed the sub-cellular localization of the overexpressed GFP-MPRIP fragments F1 and F2. Fragment F1 localizing to (**A**) stress fibers and (**B**) to nucleus. Fragment F2 shows (**C**) globular and fibrous structures in the cytoplasm but not in the (**D**) nucleus. Scale bars represent 10 µm.

**Figure 11 cells-10-00848-f011:**
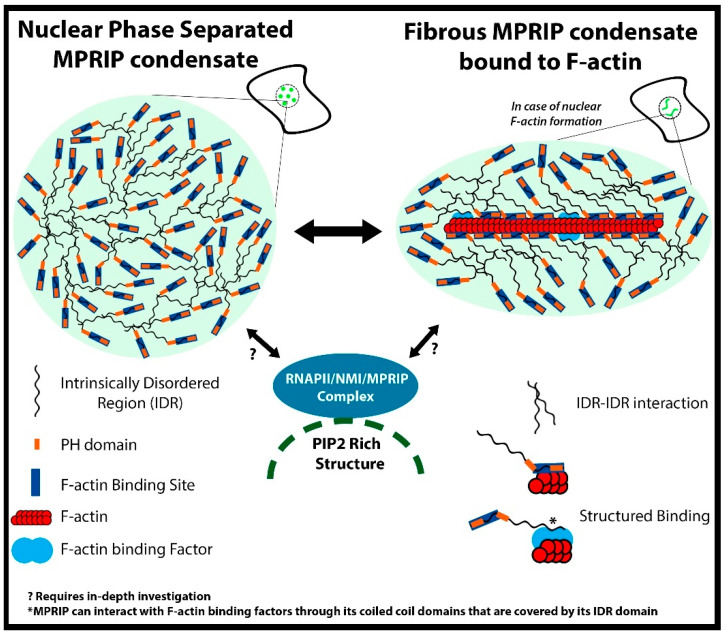
Illustration of the hypothetical model of interplay between MPRIP phase separation and its interactors. RNAPII/NMI/MPRIP complex localizes in the proximity of nuclear PIP2-rich structures. The MPRIP protein contains two PH domains, an F-actin binding site and C-terminal IDR. MPRIP is able to form phase-separated condensates through IDR interactions. These condensates can form fibers when associated with nuclear F-actin. (*) This association is mediated either by intrinsic MPRIP F-actin-binding site or by other F-actin binding factors. (?) It remains unclear how MPRIP/RNAPII/NM1/PIP2 complex formation depends on the phase separation, actin polymerization or association with PIP2-rich nuclear structures. It also needs further clarification as to how these processes orchestrate the MPRIP-mediated regulation of RNAPII activity and the consequent transcriptional output of the cell.

## Data Availability

Data is contained within the article or Appendix A.

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
