# Peer review of "The F-Actin-Binding MPRIP Forms Phase-Separated Condensates and Associates with PI(4,5)P2 and Active RNA Polymerase II in the Cell Nucleus"

_cells, 2021, doi:10.3390/cells10040848_

Round 1

Reviewer 1 Report

The authors investigate localization and interactions of MPRIP in the cell nucleus. They use immunoprecipitation, pull-down assay, confocal microscopy and other advanced imaging techniques (STED, FRAP) to reveal molecular interactions of nuclear MPRIP. The results show that MPRIP interacts with PIP2 and forms complex with RNAPII and MYO1C in the nucleus. Overexpression of MPRIP leads to the formation of phase-separated condensates and fibrous structures. MPRIP condensates show dynamic liquid-like properties, and are able to fuse, divide and transform into fibers. The authors also showed that the C-terminal part is responsible for the phase separation of MPRIP. They propose that MPRIP might contribute to the regulation of RNAPII-dependent transcription.  

Specific comments

Introduction

In line 36, the citation number [11] seems to be inappropriate.

Material and Methods

In line 65-67, at the description of stable cell lines, the authors should refer to the supplementary Figure S6, where these cells are investigated.

In the subsection 2.5 (starting in line 111) the buffer used for the resuspension of the nuclear pellet is omitted. Since total nuclear lysates are prepared, the title ‘Nuclear extract fractionation’ sounds inappropriate. The authors mention here also that these nuclear lysates were used for pull-down experiments, but neither the protocol of the pull-down assay nor the source of the PIP-coated agarose resin are mentioned in the Materials and Methods section.  

There are two different citations mentioned for nuclear lysate preparation (line 113 and line 124). Is there a need for both?

Results

The subsections are incorrectly numbered.

The authors have performed their experiments either on U2OS or HeLa cells. What was the reason for choosing these two cell lines?

In Fig. 1. a strong band of MPRIP is detected in the nuclear fraction of HeLa cells by Western blotting. Have the authors checked MPRIP localization in HeLa cells with IF labelling, too? Does that also show a predominant nuclear localization of MPRIP in these cells?

The type of the cell lysates applied for the pull-down assay in Fig. 2. and immunoprecipitation in Fig. 4. are not mentioned in the respective figure legends.

In line 252, there is a reference to Fig. 1A instead of Fig. 5A.

The last figure (at line 455) has no number, legend or any description in the main text.

Additional questions

Why did the authors choose a phospho-specific RNAP antibody for immunoprecipitation assay? It should be commented in the text, too.

How can the authors interpret the predominant nuclear localization of GFP-MPRIP (Fig. 5) compared to the subcellular distribution of endogenous MPRIP?

The hypothesis that MPRIP contributes to the regulation of RNAPII transcription could be strengthen by checking the RNAPII-dependent transcription level in MPRIP overexpressing or depleted cells.

Author Response

Specific comments

Introduction

In line 36, the citation number [11] seems to be inappropriate.

  • Line 35 (corrected): Reference number [11] was changed to [12] as it was initially intended.

Material and Methods

In line 65-67, at the description of stable cell lines, the authors should refer to the supplementary Figure S6, where these cells are investigated.

  • In lines 65-68 were corrected and reference to Figure S6 was added.

In the subsection 2.5 (starting in line 111) the buffer used for the resuspension of the nuclear pellet is omitted. Since total nuclear lysates are prepared, the title ‘Nuclear extract fractionation’ sounds inappropriate. The authors mention here also that these nuclear lysates were used for pull-down experiments, but neither the protocol of the pull-down assay nor the source of the PIP-coated agarose resin are mentioned in the Materials and Methods section.  

  • Line 106: The name of the subsection 2.6 . was changes to “Nuclear extraction and Pull-down assay”.
  • Lines 111-112 the composition of the buffer composition was included.
  • Lines 118-127 the protocol for the Pull down added including the source of the PIP-coated agarose resin used in these experiments.

There are two different citations mentioned for nuclear lysate preparation (line 113 and line 124). Is there a need for both?

  • Line 129 – This was a mistake and only citation number 34 was left in the text.

Results

The subsections are incorrectly numbered.

  • The subsections were corrected in the whole manuscript.

The authors have performed their experiments either on U2OS or HeLa cells. What was the reason for choosing these two cell lines?

  • Human U2OS cell line originates from osteosarcoma and represents a “golden standard” for studies on nuclear architecture (live-cell imaging, immunofluorescence, EM, etc.) due to the convenient manipulations enabled by good transfection efficacy and size of their nucleus which varies around ~15x20 µm. However, U2OS is an adherent cell line. Therefore, to obtain the high cell amount needed for biochemical experiments and proteomics our laboratory typically uses suspension HeLa cells which can be easily cultivated and thus enable to obtain incomparably higher protein yields than from U2OS cells. To overcome this limitation, our laboratory uses HeLa cells which can be easily cultivated in suspension cultures and thus enable incomparably higher protein yields than from U2OS cells.

In Fig. 1. a strong band of MPRIP is detected in the nuclear fraction of HeLa cells by Western blotting. Have the authors checked MPRIP localization in HeLa cells with IF labelling, too? Does that also show a predominant nuclear localization of MPRIP in these cells?

  • As stated above, in our laboratory we use both types of the cell lines for different experimental purposes. These cell types were carefully chosen for their similarities in respect to nuclear architecture, phosphoinositides and transcriptional rate. Both cell lines were tested for the presence of all nuclear factors important for our research including MPRIP. Until now, we did not observe any major differences in the composition of their nuclear lysates (validations IF, EM, WB, and MS).

The type of the cell lysates applied for the pull-down assay in Fig. 2. and immunoprecipitation in Fig. 4. are not mentioned in the respective figure legends?

  • Lines 209 and 239-240 - Cell types are added into both figure legends.

In line 252, there is a reference to Fig. 1A instead of Fig. 5A.

  • Line 247 – corrected to Fig. 5A reference

The last figure (at line 455) has no number, legend or any description in the main text.

  • Lines 433-448 - Figure number, legend and description added.
  • Line 430 – reference to figure 11 added to the main text.

Additional questions

Why did the authors choose a phospho-specific RNAP antibody for immunoprecipitation assay? It should be commented in the text, too.

  • In lines 230-236 – description of the Ser5 in respect to nuclear PIP2. Phosphorylation of Serine 5 within CTD domain of Pol2 is connected with the initiation stage of transcription which precedes to an active transcriptional burst. We have previously shown that Pol2 phosphorylated at Ser5 associates with nuclear PIP2 at surface of NLIs structures [Ref 12].
  • In line 235 -236 - we added that phospho Ser5 refers to an active Pol2.

How can the authors interpret the predominant nuclear localization of GFP-MPRIP (Fig. 5) compared to the subcellular distribution of endogenous MPRIP?

  • This is indeed an interesting question which deserves further exploration. We plan to address this question in the following project. The MPRIP contains a predicted NLS (supp. fig. 1) in its N-terminal region which directs the protein into the nucleus (Fig. 10). The computational predictions suggest that this NLS is very strong. The high expression level of MPRIP triggers the phase separation within the nucleus (Fig. 5). The phase separation of full-length MPRIP within the cytosol was not observed. Whereas, the overexpression of C-terminal part containing IDR leads to phase separation in cytosol only. Due to a lack of data, we can only speculate that the binding of MPRIP to cytosolic F-actin keeps the level of nuclear MPRIP relatively low. When we artificially increase the MPRIP concentration by overexpression, the effect of F-actin sequestration is overridden by the high amount of the MPRIP protein. As a result, the MPRIP goes into the nucleus where it accumulates in high amounts reaching the critical concertation that leads to phase separation.

The hypothesis that MPRIP contributes to the regulation of RNAPII transcription could be strengthen by checking the RNAPII-dependent transcription level in MPRIP overexpressing or depleted cells.

We agree that this is an important point which we are currently investigating and will be part of a study describing the molecular action of MPRIP protein in respect to RNAPII transcription.

Reviewer 2 Report

The authors aimed to examine how the F-actin-binding protein MPRIP exists and localizes in cell nucleus. Using fluorescence STED microscopy and FRAP they found that MPRIP localizes to PIP2-containing nuclear structures, forms complex with RNAPII and MYO1C and undergoes LLPS. The experiments were well conducted.  However, I raised only a few comments.  

They overexpressed MPRIP. This may induce artefacts in the cell. Please comment in the text.

In the model shown in the end, they show only actin and MPRIP.  RNAPII, PIP2 and NM1 (MYO1C) should be included for helpful and better understanding.

Myosin IC (NM1) is a motor protein and binds to F-actin to move or relocate the complex.   But the authors say that NM1 is also a transcription factor.  It is very confusing. Do they have any evidence that NM1 bound to RNAPII affects transcription rate as a transcription switch? Please comment in the text.

Please check the references. Ref 1 and 63 are the same.

Author Response

They overexpressed MPRIP. This may induce artefacts in the cell. Please comment in the text.

  • The condensates formed in the cell nucleus by the MPRIP overexpression are definitely non-physiological structures. The studies aiming to describe the properties of proteins which have the ability to phase separate standardly use similar systems of protein overexpression, where the high concentration of particular protein trigger its condensation and thus enables the subsequent experimentations. In the nuclear environment the formation of bigger condensates is strictly regulated and small microcondensates are preferentially formed. The nucleus is thus a multiphase environment with the properties of microemultions which possess the crucial intrinsic characteristics defined by the physics of soft matters (Hilbert et al., 2021; PMID: 33649325). Thus the phase separation represents the important phenomenon which needs to be extensively studied in order to understand the spatiotemporal orchestration of such processes as RNA transcription.

In the model shown in the end, they show only actin and MPRIP.  RNAPII, PIP2 and NM1 (MYO1C) should be included for helpful and better understanding.

  • We thank this reviewer for the suggestion. We implemented recomended modifications in our model (Figure 11).

Myosin IC (NM1) is a motor protein and binds to F-actin to move or relocate the complex.   But the authors say that NM1 is also a transcription factor.  It is very confusing. Do they have any evidence that NM1 bound to RNAPII affects transcription rate as a transcription switch? Please comment in the text.

  • The MYO1C is an unconventional monomeric myosin motor protein. Its nuclear isoform called NMI contains a unique N-terminal region. Interestingly, this isoform has been previously shown to act as a noncanonical transcription factor important for RNA polymerase I and II [ref 12, 60, 66].

Please check the references. Ref 1 and 63 are the same.

  • Corrected: Ref number 63 is taken out

Round 2

Reviewer 1 Report

The publication can be accepted in the present form.

Reviewer 2 Report

The authors aimed to examine how the F-actin-binding protein MPRIP exists and localizes in cell nucleus. Using fluorescence STED microscopy and FRAP they found that MPRIP localizes to PIP2-containing nuclear structures, forms complex with RNAPII and MYO1C and undergoes LLPS. The experiments were well conducted.  However, I raised only a few comments.  

They overexpressed MPRIP. This may induce artefacts in the cell. Please comment in the text.

The answer is satisfactory, although in the revised form, they did not describe it in the text.

In the model shown in the end, they show only actin and MPRIP.  RNAPII, PIP2 and NM1 (MYO1C) should be included for helpful and better understanding.

The authors revised the model. It is satisfactory.

Myosin IC (NM1) is a motor protein and binds to F-actin to move or relocate the complex.   But the authors say that NM1 is also a transcription factor.  It is very confusing. Do they have any evidence that NM1 bound to RNAPII affects transcription rate as a transcription switch? Please comment in the text.

They cited previous literatures as references to explain dual function of NM1.

Please check the references. Ref 1 and 63 are the same.

The authors corrected it.